# Multi-GNSS Large Areas PPP-RTK Performance During Ionosphere Anomaly Periods

**DOI:** 10.3390/s25072200

**Published:** 2025-03-31

**Authors:** Zhu Wang, Guangbin Yang, Rui Huang, Man Li, Menglan Zhu

**Affiliations:** 1School of Geography & Environmental Science, Guizhou Normal University, Guiyang 550001, China; wangzhu_88@msn.com (Z.W.); liman4515@163.com (M.L.); 2Real Estate Registration Center, Department of Natural Resources of Guizhou, Guiyang 550004, China; 3GIS Center, Power China Guiyang Engineering Corporation Limited, Guiyang 550081, China; huangrui_gyy@powerchina.cn (R.H.); zhumenglan_gyy@powerchina.cn (M.Z.)

**Keywords:** PPP-RTK, ambiguity resolution, ionospheric scintillation, service performance

## Abstract

Precise Point Positioning with real-time kinematic (PPP-RTK) technology, which relies on Global Navigation Satellite Systems (GNSS), encounters difficulties in achieving high-precision and rapid convergence during ionospheric active conditions such as those occurring in thunderstorms. Most existing research on PPP-RTK has primarily focused on calm ionospheric conditions, with limited analysis of its performance under ionospheric anomalies. This study analyzes 13-day data collected from 305 Australian stations, encompassing both ionospheric anomalies (from 10 to 13 May 2024) and calm periods. We evaluated the residuals of uncalibrated phase delay (UPD), the accuracy of atmospheric modeling, as well as the positioning accuracy and convergence time of PPP-RTK. The results reveal that during ionospheric anomalies, compared to calm conditions, the accuracy of wide-lane and narrow-lane UPDs decreases by 2.4% and 1.4%, respectively. Meanwhile, the accuracy of estimated ionospheric and tropospheric delays deteriorates by 167.1% and 17.3%, respectively. In terms of PPP-RTK services, for the horizontal component, the convergence times increase by 25.0%, 44.4%, and 55.6% for the GPS-only, GPS + Galileo, and GPS + Galileo + BDS solutions, respectively. For the vertical component, the increases are 56.9%, 81.6%, and 87.2%, respectively. Regarding the positioning accuracies, for the horizontal component, they decline by 5.5%, 7.4%, and 10.4% for the GPS-only, GPS + Galileo, and GPS + Galileo + BDS solutions, respectively. For the vertical component, the declines are 11.8%, 13.0%, and 18.5%, respectively. This indicates that ionospheric anomalies significantly disrupt PPP-RTK services, mainly due to the degradation of ionospheric delay estimates, which directly affects positioning results. Although the ionosphere can lead to significant degradation in positioning performance, the positioning performance can still be substantially improved with an increase in the number of satellites. This study thus offers new insights into the performance of PPP-RTK during ionospheric active conditions.

## 1. Introduction

Precise Point Positioning (PPP) relying on the Global Navigation Satellite System (GNSS) has become a mature technology after decades of development and is widely applied in fields such as geoscience, autonomous driving, precision agriculture, and geological disaster monitoring. However, atmospheric conditions can significantly impact GNSS signals, affecting both the accuracy and timeliness of real-time PPP. Disturbances in the ionosphere, such as those caused by solar activity or geomagnetic storms, can cause delays in signal propagation, increasing errors in positioning.

Atmospheric delay can be divided into tropospheric delay and ionospheric delay. Among them, the ionospheric delay is generally tens or hundreds of times that of the tropospheric delay, and its accuracy directly affects positioning accuracy and convergence speed. To solve ionospheric delay in positioning, the legacy PPP technology adopts the ionosphere-free (IF) combination or estimates as a parameter in undifferenced and uncombined (UDUC) modes [1,2]. However, the ionosphere parameters and ambiguity parameters require a relatively long time to separate. With the advancement of real-time orbit, clock, and ambiguity resolution (AR) technologies [3,4,5,6,7,8,9], PPP with external ionospheric and tropospheric delays has found extensive application in numerous scientific research endeavors and civilian applications [10,11,12]. This enables rapid resolution of initial carrier-phase ambiguities within a few mins, effectively implementing PPP with Real-Time Kinematic (PPP-RTK) [13].

PPP–RTK combines the benefits of both PPP and RTK methods. It enhances PPP by providing users with satellite phase biases, facilitating single-receiver ambiguity resolution, and achieving RTK-like positioning performance. PPP–RTK is more flexible than RTK, as it utilizes the State Space Representation (SSR) approach, which is easier to broadcast and covers larger areas with less data volume. It can be implemented over large regions with sparse reference station networks and is not limited by the communication restrictions of ground networks. Satellite orbit, clock offsets, and uncalibrated phase delay (UPD) products can be broadcast through satellite signals or network communication. Additionally, high-precision atmospheric information can be provided through various enhancement methods, such as low-orbit satellites or the Internet [14].

Recently, some researchers were the first to propose the Uncombined and Un-differenced Ionospheric-Free Combined PPP-RTK (UDUC PPP-RTK) by selecting a distinct clock datum [15,16], while Zhang et al. suggested adopting a common clock datum [17]. The common clock model allows the estimation of receiver and satellite biases instead of being absorbed into distinct clocks. When the biases are properly constrained, it provides a stronger model. Subsequently, researchers extended the common clock undifferenced and uncombined PPP–RTK from dual-frequency GPS cases to multi-frequency and multi-GNSS scenarios [5,18]. Imposing various constraints on atmospheric delays has led to different PPP–RTK variations suitable for small-scale, medium-scale, and large-scale networks [19,20,21,22,23,24]. Even in the specific case where the network consists of only one receiver, researchers have introduced the concept of single-station PPP–RTK [25]. For different conditions, Geng et al. evaluated the performance of all-frequency GPS/Galileo/BDS PPP-RTK in challenging GNSS environments, including demonstrating its positioning accuracy and ambiguity fixing rate in static and vehicle-mounted scenarios through experiments, but the case of an abnormal ionosphere was not considered [26]. Li et al. use factor graph-based PPP-RTK in urban environments to further improve the accuracy and robustness of PPP-RTK solutions [27]. Furthermore, for the various proposed PPP-RTK models and improved algorithms, numerous studies have conducted evaluations and analyses on the performance of PPP-RTK [28].

However, for PPP-RTK, the most crucial aspect is the accuracy of ionospheric delay products. At present, many researchers have conducted studies on the ionosphere of PPP-RTK, indicating solar activities are related to ionospheric events. The evolution of the ionosphere is significantly modulated by the level of solar radiation and solar activity events through complex mechanisms [29]. Existing studies have shown that ionospheric anomalies can have a significant impact on the availability of GNSS services and interfere with the availability of data and have been analyzed and evaluated in mid-latitude and low-latitude regions [30,31,32]. To achieve better performance under different ionospheric conditions, Li et al. introduced cross-verified ionospheric uncertainty to achieve more appropriate ionospheric constraints and adapt to different ionospheric environments [30]. However, in the case of low elevation angles or abnormal ionospheric conditions, the existing calculation methods for ionospheric delay and uncertainty of PPP-RTK still cannot obtain accurate enhanced information.

Currently, research on PPP-RTK has primarily focused on evaluating and analyzing positioning performance during periods of ionospheric calm. The ionosphere is highly variable, especially during periods of high solar activity, geomagnetic storms, or even during sunrise and sunset. Studies limited to calm conditions fail to account for the disturbances and variations that can significantly impact GNSS signal propagation, leading to errors in positioning. Real-world applications of PPP-RTK must consider these challenging conditions to ensure reliable performance. GNSS applications that use PPP-RTK for precise positioning are often deployed in diverse environmental conditions, not just during calm ionospheric periods. By focusing solely on calm conditions, studies overlook the robustness of the system in adverse scenarios, where performance might degrade or require additional mitigation strategies. Few studies have focused on the performance of PPP-RTK and atmospheric delay differences during periods of ionospheric activity, especially during abnormal solar activity.

In May 2024, NASA and NOAA reported that a series of intense solar flares and coronal mass ejections (CMEs) sent vast clouds of charged particles and magnetic fields toward Earth. This resulted in the most powerful geomagnetic storm in two decades, potentially offering one of the most spectacular aurora displays seen in the last 500 years [33]. During periods of increased solar activity, such as solar maximum, the Sun emits higher levels of solar radiation, which intensifies ionospheric activity. This results in more frequent and severe ionospheric disturbances, including ionospheric scintillation and Total Electron Content (TEC) fluctuations. Since PPP-RTK relies on precise measurements of signal delays through the atmosphere, understanding how these disturbances impact signal quality and accuracy is critical for improving performance in real-world conditions. This particular period of solar activity offers a unique opportunity for PPP-RTK performance during ionosphere anomaly. To address that gap, the period of abnormal solar activity from 10 to 13 May 2024 was selected to specifically explore positioning and atmospheric estimation performance under both calm and active ionospheric conditions.

The structure of this paper is as follows. First, we introduce the PPP-RTK, UPD estimation, and precise atmospheric delay deriving methods. Next, the observation data and detailed processing strategies of the experiment are presented. Finally, the results and analysis of estimated UPDs, atmospheric delay differences, and PPP-RTK performance are compared and discussed, followed by the conclusions.

## 2. Materials and Methods

This study begins with GNSS observation equations to explain the methods of PPP-RTK used to derive atmospheric delay and UPD estimation on the server side. Then, we present the positioning with ambiguity resolution and atmospheric delay constraint on the user side.

### 2.1. GNSS Observation Equations

The raw GNSS observations in the unit of length from satellite s to receiver r take the form of(1)$$\left\{\begin{array}{c} P_{r,f}^{s}={\vec{\rho }}_{r}^{s}+c\left({dt}_{r}-{dt}^{s}\right)+T_{r}^{s}+\gamma_{f}I_{r,1}^{s}+c\left(d_{r,f}+d_{f}^{s}\right)+\varepsilon_{P,f}\\ L_{r,f}^{s}={\vec{\rho }}_{r}^{s}+c\left({dt}_{r}-{dt}^{s}\right)+T_{r}^{s}-\gamma_{f}I_{r,1}^{s}+\lambda_{f}N_{f}+c\left(b_{r,f}+b_{f}^{s}\right)+\varepsilon_{L,f} \end{array}\right.$$
where superscript s, subscript r, and f are satellite, receiver, and frequency band, respectively; $P_{r,f}^{s}$ and $L_{r,f}^{s}$ are the pseudo-range and the carrier-phase observations with all necessary corrections listed in Table 1; c is the light speed, ${dt}_{r}$ and ${dt}^{s}$ are receiver and satellite clock offsets; $T_{r}^{s}$ is the slant troposphere delay; $I_{r,1}^{s}$ is ionospheric delay along the line-of-sight from satellite to receiver at the frequency L1 and $\gamma_{f}=f_{1}^{2}/f_{f}^{2}$; $\lambda_{f}$ is wavelength; $N_{f}$ is phase ambiguity; $b_{r,f}$ and $b_{f}^{s}$ are code and phase biases in receiver and satellite sides, respectively; and $\varepsilon_{P,f}$ and $\varepsilon_{L,f}$ are code and phase measurement noise including the multipath effect, respectively.

Typically, PPP technology relies on external high-precision products. In our study, we obtained high-precision real-time orbit and clock products from Centre national d’études spatiales (CNES). By applying these corrections, along with products for antenna calibration from the International GNSS Service (IGS) and pseudorange signal bias correction from the Chinese Academy of Sciences (CAS), the unknown parameters are reduced to tropospheric and ionospheric delays, coordinates, ambiguities, and receiver clock offsets. In PPP-RTK service, undifferenced and uncombined PPP modes are generally applied to derive the ionospheric delay of the satellite–station pair, and Equation (1) can re-parameterization as,(2)$${∆P}_{r,f}^{s}={\vec{e}}_{r}^{s}\cdot ∆\vec{x}+c\cdot \hat{∆}t_{r}+\gamma_{f}\cdot {\hat{I}}_{r,1}^{s}+m^{s}\cdot {ZWD}_{r}+\varepsilon_{P,f}$$(3)$${∆L}_{r,f}^{s}={\vec{e}}_{r}^{s}\cdot ∆\vec{x}+c\cdot \hat{∆}t_{r}+\gamma_{f}\cdot {\hat{I}}_{r,1}^{s}+m^{s}\cdot {ZWD}_{r}+\lambda_{f}^{s}\cdot {\hat{N}}_{r,f}^{s}+\varepsilon_{L,f}$$
where ${∆P}_{r,f}^{s}$ and ${∆L}_{r,f}^{s}$ are the pseudo-range and the carrier-phase observed-minus-computed value; ${\vec{e}}_{r}^{s}$ is the unit vector from satellite to receiver; $∆\vec{x}=[∆x∆y∆z]$ is the receiver coordinates increments vector; $\hat{∆}t_{r}$ is receiver end clock absorbed pseudo-range biases; ${\hat{I}}_{r,1}^{s}$ is slant ionospheric delay including pseudo-range biases; $m^{s}$ is the mapping function of tropospheric zenith wet delay (ZWD) ${ZWD}_{r}$; and ${\hat{N}}_{r,f}^{s}$ is phase ambiguities, which have been changed due to the re-parameterization.

For GNSS high-precision positioning, the most critical aspect is achieving accurate ambiguity resolution. In this study, we focus on dual-frequency observations, and therefore, we first estimate the wide-lane (WL) and narrow-lane (NL) uncalibrated phase delay (UPD) derived from dual-frequency ionosphere-free combinations, which are essential for ambiguity resolution. To fix the ambiguity, the float phase ambiguities’ detailed forms are given as follows:(4)$${\hat{N}}_{r,f}^{s}=N_{r,f}^{s}+\left[c/\left(\lambda_{f}^{s}\left(\gamma_{2}-1\right)\right)\right]\times \left[\left(\gamma_{f}-1\right)\left(d_{r,2}+d_{2}^{s}\right)-\left(\gamma_{f}+\gamma_{2}\right)\left(d_{r,1}+d_{1}^{s}\right)\right]$$

To resolve the integer ambiguity in PPP, the UPD must be calibrated for the carrier-phase ambiguities. We are following Ge, et al. to estimate WL and NL UPDs in ambiguity resolution [34]. The WL UPDs are calculated using the Melbourne–Wübbena (MW) combination, while the NL UPDs are estimated based on the NL ambiguities derived from IF ambiguities with fixed WL integers. Their usability in UDUC-PPP has been demonstrated [23,35]. The relationships among *IF*, L1, L2, *WL*, and *NL* ambiguities can be expressed as follows to illustrate the principle for UPD estimation on the server side and using UPDs for ambiguity resolution on both the server side and user side [35].(5)$${\bar{N}}_{r,IF}^{s}=\frac{f_{1}^{2}}{f_{1}^{2}-f_{2}^{2}}{\bar{N}}_{r,1}^{s}-\frac{f_{1}f_{2}}{f_{1}^{2}-f_{2}^{2}}{\bar{N}}_{r,2}^{s}=\frac{f_{1}}{f_{1}+f_{2}}{\bar{N}}_{r,NL}^{s}-\frac{f_{1}f_{2}}{f_{1}^{2}-f_{2}^{2}}N_{r,WL}^{s}$$

With(6)$${\bar{N}}_{r,WL}^{s}=N_{r,WL}^{s}+d_{r,WL}-d_{WL}^{s}\;{\bar{N}}_{r,NL}^{s}=N_{r,NL}^{s}+d_{r,NL}-d_{NL}^{s}$$
where $N_{r,WL}^{s}$, ${\bar{N}}_{r,WL}^{s}$, $N_{r,NL}^{s}$, and ${\bar{N}}_{r,NL}^{s}$ are *WL* integer and float, and *NL* integer and float ambiguities, respectively; $d_{r,WL}$, $d_{WL}^{s}$, $d_{r,NL}$, and $d_{NL}^{s}$ are *WL* and *NL* UPDs in receiver and satellite side, respectively. From Equation (5) either the estimated *IF* ambiguity or L1 and L2 ambiguities can be decomposed into the corresponding *WL* and *NL* ambiguities. If the float estimates of all the ambiguities are obtained, the UPD can be estimated according to Equation (6) for *WL* and *NL*, respectively. Reversely, with achieved *WL* and *NL* UPD products, the related ambiguities can be fixed at a single receiver. Therefore, the remaining parameters to be estimated are(7)$$X=(\Delta r^{T},{ZWD}_{r},\hat{∆}t_{r},(ISB^{Sys}{)}^{T},(IFB^{s}{)}^{T},({\hat{I}}_{r,1}^{s}{)}^{T}{)}^{T}$$

The Kalman filter is employed for the parameter estimation. Tropospheric ${ZWD}_{r}$ and ionospheric slant delay ${\hat{I}}_{r,1}^{s}$ are estimated as random walk process noise, respectively. The receiver clock $\hat{∆}t_{r}$ is estimated as epoch-wise white noise and the inter-system bias $ISB^{Sys}$ and inter-frequency bias $IFB^{s}$ parameters are estimated as constant values over time. $ISB^{Sys}$ are applied for Code Division Multiple Access (CDMA) satellite, i.e., BDS and Galileo constellations, while and $IFB^{s}$ are applied to GLONASS Frequency-division multiple access (FDMA) mode satellites.

For each reference station, the PPP-AR mode with fixed coordinates is performed to derive the tropospheric ZWD and slant ionospheric delays of all satellites. Equation (2) can be further transformed as follows:(8)$$l_{r,f}^{s}={\hat{t}}_{r}+m_{r,w}^{s}T_{r,w}-\gamma_{f}\cdot {\hat{I}}_{r,1}^{s}+\varepsilon_{l,f}^{s}$$

The tropospheric ZWD is estimated using the random walk process with a power spectral density of 3 mm/sqrt(h), and slant ionospheric delay is estimated as white noise. The IDW algorithm is employed to provide interpolated tropospheric and ionospheric delay products, with ionospheric uncertainties calculated using the cross-validation method.

### 2.2. Positioning with Atmospheric Accuracy Constraint

On the server side, tropospheric and ionospheric delays are estimated and derived from each reference station and used as inputs to calculate the interpolated corrections. The interpolated ZWD and slant ionospheric delays are then added as virtual observations to enhance the PPP-AR solution on the user side.(9)$$\delta_{s}=f_{s}-F_{s},{\;\;\sigma }^{2}$$
where $\delta_{s}$ represent the differences between the nearby reference station interpolated values and the PPP-AR derived values with the uncertainty $\sigma^{2}$, $f_{s}$ is an interpolated value from nearby reference stations, $F_{s}$ is the PPP-AR derived value [36].

**Table 1 sensors-25-02200-t001:** Data processing strategies.

Items	Strategy	
	Service-Side	User-Side
Data	GPS + Galileo + BDS double-frequency with 30 s interval	
Elevation cutoff	7°	
Satellite orbit and clock	GFZ real-time streams products [37]	
Tropospheric ZWD delay	Saastamoinen + VMF3 + GPT3 + random work processing [38]	Three nearby stations interpolated
Tropospheric ZHD delay	Saastamoinen + VMF3 + GPT3	
Ionospheric delay	Estimated as white noise	Three nearby stations interpolated
Satellite and receiver antenna	Igs20.atx	
Phase wind-up	Corrected [39]	
Phase ambiguity	WL + NL using LAMBDA to fix [40]	
Station displacement	Solid earth tides, ocean tides, and pole tide displacements corrected according to IERS 2010 [41]	
Differential Code Biases	CAS DSB (Differential code biases) [42]	
Parameter estimator	Kalman filter	

## 3. Observation Data and Processing Strategies

In this section, we detail the experiments, including the station distribution and the processing strategies for positioning.

### 3.1. Dataset

The study area encompasses the entire Australian region, utilizing GNSS data from the Australia National Positioning Infrastructure reference networks within this region (https://ga-gnss-data-rinex-v1.s3.amazonaws.com/index.html#public/, accessed on 1 May 2024). To compare the service performance of PPP-RTK under different ionosphere conditions, data are selected for ten days during the ionospheric calm period from 1 to 10 May 2024, and three days during the period of ionosphere activity from 10 to 13 May 2024. This dataset encompasses signals from GPS, Galileo, and BDS-3 satellites, and all stations can fully support GPS L1/L2, Galileo E1/E5a, GLONASS L1/L2, and BDS-3 B1/B3 frequencies.

The ionospheric delay Rate of TEC Indexes (ROTI) for these periods are presented in Figure 1. The ionosphere ROTI is a key parameter used to describe and monitor the state of the ionosphere, specifically the electron density [43]. It plays a crucial role in understanding the ionosphere’s response to solar activity and other external factors. Generally, the ROTI remains below 0.2, and it can be considered as in the quiet period. However, as observed in the ionospheric activity conditions (as shown in the figure below), a significant increase in ionospheric activity becomes evident after 11 May. Therefore, in this study, we define ionospheric conditions with ROTI values exceeding 0.2 as indicative of an active state. Firstly, high-precision ionospheric delays are first extracted through PPP-AR technology using GNSS observation on all stations, followed by the ROTI calculation steps to determine the ROTI results for each station [44]. To demonstrate regional ROTI characteristics, we take the averaged ROTI results from the densest station cluster in southeastern Australia as an example, statistically analyzing the temporal variations. The results were plotted at 30-min intervals for graphical presentation.

Typically, the ionospheric variability across epochs is quantified using the ROTI. However, defining explicit thresholds for activity levels based solely on ROTI remains unclear. To address this, we further introduce the Kp index as an additional metric to characterize ionospheric activity intensity. Generally, Kp indices between 0 and 3 correspond to quiet ionospheric conditions, 3–6 indicate moderate activity, and values exceeding 6 signify active ionospheric states. The ionospheric delay Kp index for these periods is seen in Figure 2. The geomagnetic three-hourly Kp index, introduced by Matzka et al. [45], is designed to measure solar particle radiation by its magnetic effects and is considered a proxy for the energy input from the solar wind to Earth.

Stations are uniformly selected across the Australian region to solve atmospheric parameters on the server side. To validate the PPP-RTK performance, 221 external stations are chosen for positioning validation. The distribution of these stations is shown in Figure 3.

From Figure 3, it is evident that the 84 stations on the service side uniformly cover the Australian region, and the remaining 221 stations are selected for positioning verification and also ensure good coverage across Australia. Due to the uneven distribution of stations, with a dense concentration in the eastern region and a sparse distribution in the western region, we restricted those we used to ensure consistent service performance evaluation of PPP-RTK. Specifically, stations were selected such that the distance between reference and rover stations is larger than 100 km.

### 3.2. Processing Strategy

In the data processing, we divided the workflow into two parts: server-side and user-side. The primary objective of the server side is to estimate UPD products and derive atmospheric delay. To ensure the accuracy and stability of atmospheric delay, ambiguity fixing is performed at each server-side station. The user side then uses the received atmospheric delay to accelerate ambiguity fixing. Additional data processing strategies are detailed in Table 1.

The WL and NL UPDs are first generated for broadcasting, and their quality is subsequently evaluated. Following this, PPP-AR is performed to determine the tropospheric ZWD and ionospheric delays at each reference station, and the differences between the calm and active conditions are analyzed. The interpolated values and uncertainties of the tropospheric and ionospheric delays are then calculated and broadcast. Finally, the PPP-RTK service is analyzed to compare positioning performance and assess the impact during ionospheric active conditions. At the user end, reference stations are used to estimate the WL and NL UPDs products via IF static mode PPP in advance. It is important to note that while UDUC-PPP can obtain ambiguity on each frequency, IF-PPP solutions are more stable due to their reduced sensitivity to ionospheric effects [46]. Thus, using WL and NL UPDs to fix the ambiguity helps deriving atmospheric delay at each reference station. The derived ionospheric and tropospheric delays are interpolated from three nearby stations and then broadcast to the user side. The uncertainty information is calculated following the methodology of [30]. Finally, PPP-RTK is performed on the user end.

It is noteworthy that orbit, clock, and UPD product estimation utilize globally distributed reference stations. This uniform station distribution ensures continuous observation of all satellites and prevents observation data from being affected by regional ionospheric anomalies. Through long-term time series analysis and multi-station data accumulation, the impact of localized anomalies on observation data can be effectively mitigated, thereby maintaining the stability and precision of the derived orbit, clock, and UPD products. However, for user-end atmospheric delays derived from adjacent reference stations, localized ionospheric anomalies can induce significant fluctuations in the estimated ionospheric delays. Compared to quiet periods, cycle slips and gross errors increase by 13.7%, subsequently introducing biases in user positioning solutions.

Furthermore, satellite observations in ionospheric active regions frequently exhibit cycle slips and outliers, leading to repeated re-convergence of ambiguities for affected satellites. To ensure positioning reliability, our strategy incorporates partial ambiguity resolution combined with an iterative outlier exclusion process for anomalous satellites. This approach effectively maintains the robustness of positioning solutions while addressing challenges posed by ionospheric disturbances.

## 4. Experimental Validation

In this section, the WL and NL UPD products are analyzed in advance. The UPD products are estimated using the selected server end stations. High-precision tropospheric and ionospheric delays are derived by resolving the ambiguities at all service stations. Finally, the positioning accuracy was verified across all user stations.

### 4.1. UPD Estimation

The WL and NL UPD products of GPS, Galileo, and BDS satellites are estimated in advance. We present DOY 130 (ionosphere calm) and 132 (ionosphere active) calculated UPD products. The WL and NL UPDs are estimated every 30 s, i.e., each epoch.

It can be seen from Figure 4 that the GPS, Galileo, and BDS satellite WL UPDs for 130 days all show stable sequences, with STD values of 0.061, 0.063, and 0.056 cycles, respectively. Compared with the stable ionospheric conditions for 130 DOY, the ionosphere at 132 days is affected by abnormal solar activities, and the results of UPD also show larger fluctuations. The STDs of GPS, Galileo, and BDS reach 0.063, 0.081, and 0.066 cycles, respectively. It can be noted that when solar activities are abnormal, the activity of the ionosphere will give rise to a greater fluctuation in the solution of ambiguous parameters, leading to poorer stability of the estimated UPD products. Conversely, the GPS, Galileo, and BDS satellite NL UPDs for 130 DOY present stable sequences, with STD values of 0.035, 0.034, and 0.040 cycles, respectively. The NL UPD results on the 132 DOY also show large fluctuations, with the STDs of GPS, Galileo, and BDS satellites reaching 0.047, 0.041, and 0.048 cycles, respectively. Additionally, the two-day results presented also include time periods when parts of the ionosphere are moderately active. This has minimal impact on UPD estimation, with no significant differences observed. However, minor fluctuations or small jumps in UPD estimation do occur during more active ionospheric periods.

We also conducted a simultaneous analysis of fixed residuals for WL and NL UPD across the three systems (Figure 5). The WL fixed residuals on 130 DOY exhibited strong stability, with GPS, Galileo, and BDS satellites showing residuals within ±0.15 cycles at 89.3%, 85.0%, and 92.2%, respectively. On 132 DOY, these percentages slightly decreased to 87.7%, 82.3%, and 90.2%, indicating reductions of approximately 1.7%, 3.1%, and 2.2%, respectively, compared to around 130 DOY. For NL, the reductions were 1.4%, 1.5%, and 1.2%, respectively. In terms of ±0.25 cycles, the impact of active ionospheric conditions on WL UPD estimates for GPS, Galileo, and BDS was 1.7%, 3.2%, and 0.9%, while for NL, the impact was 1.0%, 1.1%, and 0.5%, respectively. This highlights that during active ionospheric periods, UPD estimation is affected, with a slightly greater influence observed on WL UPD compared to NL.

### 4.2. Atmospheric Delay Performance Evaluation

In this section, we present the PPP-AR-derived tropospheric and ionospheric delays under ionospheric calm conditions (DOY 130) and active conditions (DOY 132). We select the CNDO station as a representative example to illustrate the differences and impacts between ionosphere calm and active conditions. The differences are computed between the PPP-AR-derived values from the user station and those interpolated from three nearby stations. We begin by presenting the ionospheric delays under various conditions in Figure 6.

It can be seen from the above figure that at the CNDO station, the ionospheric delays exhibit stable results during calm periods, with all differences between user-end PPP-AR-derived values and nearby reference stations interpolated being less than 0.3 m. However, during active ionospheric periods, differences can exceed 1 m. For GPS, Galileo, and BDS systems, the STDs during calm ionospheric conditions are approximately 5.2 cm, 4.7 cm, and 4.6 cm, respectively. During active ionospheric conditions, these values can increase significantly to 12.8 cm, 12.1 cm, and 13.9 cm, respectively. This represents an increase in residuals compared to calm periods by 145.4%, 157.1%, and 200.3%, respectively.

Additionally, we present differences in tropospheric ZWD under different ionospheric condition solutions in the same periods (Figure 6).

It is evident that during periods of ionospheric anomalies, tropospheric ZWD estimations are also affected. Compared to the 1.04 cm accuracy during the calm period on DOY 130, the estimated tropospheric fluctuation increased to 1.22 cm on DOY 132 during ionospheric anomalies, representing a 17.3% increase. This illustrates how those abnormal ionospheric fluctuations impact PPP-AR solutions, leading to significant discrepancies in both ionospheric and tropospheric delay estimation. During abnormal ionospheric conditions, these errors may become more intricate and pronounced, resulting in substantial deviations in ionospheric and tropospheric delays. For instance, variations in ionospheric delay can introduce errors in satellite signal propagation time calculations, thereby affecting positioning accuracy and leading to inaccuracies in tropospheric parameter estimates. We also conducted a statistical analysis of all station interpolated results on these days, as summarized in Table 2.

The table shows that during the ionospheric calm period, the average difference in STD is only 4.97 cm and 7.83 at ionospheric medium conditions. However, when the ionosphere condition is abnormal, the STD of the difference increases significantly to 13.86 cm, a 178.9% rise compared to the calm period. For the troposphere, due to the inaccurate estimation of the PPP solutions, the STDs during calm and abnormal periods are 1.01, 1.15, and 1.32 cm, respectively, representing an increase of 30.7% between clam and active conditions.

## 5. Positioning Validation

In this section, our analysis starts with investigating the positioning performance of PPP-AR in the ionosphere active period and the ionosphere calm period by utilizing the estimated UPD products. Then, we carry out an evaluation on the convergence time and positioning accuracy of PPP-RTK. Firstly, we present the GPS + Galileo + BDS satellites positioning result of CNDO and ARGN stations as examples on DOY 130 (ionosphere calm condition) and DOY 132 (ionosphere active condition) in Figure 7.

It can be seen from Figure 8 that there are significant differences in the positioning results during ionospheric quiet and active times at the CNDO station located in the east of Australia and the ARGN station in the western region of Australia. We select the positioning results of two hours for comparison. The ionospheric calm period is from 12:00 to 14:00 on the DOY 130, while the ionospheric active period is from 12:00 to 14:00 on the DOY 132. The sampling rate is 30 s, and the observation signals of the GPS, Galileo, and BDS systems were used.

For the CNDO station, it only takes 2 min to achieve rapid convergence and ambiguity fixing during the ionospheric quiet period. However, during the period when the ionosphere is more active, it even takes 80 min for the positioning results to show a smooth characteristic. Compared to the RMS of 1.9, 3.0, and 4.7 cm in the E, N, and U directions during the quiet period, when the ionosphere is active, the positioning accuracy decreases to 2.8, 4.6, and 8.2 cm. At the ARGN station, the convergence time is about 3 min during the ionospheric quiet period, while it takes 30 min to complete the convergence during the active period. Compared to the positioning accuracy of 2.8, 1.9, and 4.1 cm during the quiet period, the active ionosphere leads to a decrease in positioning accuracy to 4.0, 4.6, and 10.2 cm. Overall, both the eastern and western regions of Australia are affected by the active ionosphere, and the PPP-RTK positioning results and convergence time are greatly affected.

Finally, we conducted a statistical analysis of PPP-RTK positioning performance using GPS, Galileo, and BDS systems, focusing on convergence time and positioning accuracy. These metrics were separately calculated and presented. We used the static daily solution results during the ionospheric calm period as the reference coordinates. Regarding the evaluation of the PPP-RTK in the ionosphere active, medium, and calm periods with respect to convergence time and positioning accuracy, we divide the daily observations from all 221 stations (depicted in Figure 2) over a span of 13 days into 12 two-hourly sub-sessions. For 221 positioning verification stations, the 13-day continuous observations resulted in 34,476 solution arcs, with approximately 7500 arcs occurring during the ionospheric active period and 5900 periods during medium condition. Based on the obtained data, we calculate the 90th percentile positioning results for three solutions, namely GPS-only, GPS + Galileo, and GPS + Galileo + BDS-3. The convergence time is defined as the epoch when the 90th percentile positioning error drops below 10 cm. The 90th percentile result is ascertained by arranging the results in ascending order and then choosing the value at the 90th percentile to represent the convergence time and positioning accuracy.

We analyzed the average positioning results for different constellation combinations. Figure 9 illustrates that in the horizontal component, the average convergence times in the ionosphere active period for GPS-only, GPS + Galileo, and GPS + Galileo + BDS are 4.8, 2.7, and 1.8 min respectively. Under medium ionospheric conditions, the three systems exhibit convergence times of 3.9, 2.0, and 1.4 min in the horizontal component, and 4.2, 2.3, and 1.7 min in the vertical component, respectively. By contrast, in the ionosphere calm period, these convergence times are shortened to 3.6, 1.5, and 0.8 min, which means improvements of around 25.0%, 44.4%, and 55.6% with respect to the active condition, respectively. When it comes to the vertical component, for GPS-only, the convergence times in the ionosphere active and calm periods are 7.2 and 3.6 min, respectively. For GPS + Galileo, they are 4.9 and 1.5 min respectively, and for GPS + Galileo + BDS, they are 3.9 and 0.8 min respectively. On average, in the ionosphere calm period, compared to the ionosphere active period, the convergence improvements for these three solutions are approximately 56.9%, 81.6%, and 87.2% with respect to the active condition, respectively.

Furthermore, we further statistically analyzed the positioning accuracies of different systems, including GPS, GPS + Galileo, and GPS + Galileo + BDS, and presented the positioning accuracies during the active, medium, and clam periods of the ionosphere, respectively, as shown in Figure 10.

We also present the positioning accuracy results after convergence in Figure 10. It can be observed that the positioning accuracy is superior in the ionosphere calm period. For the horizontal component, the positioning accuracies for GPS-only, GPS + Galileo, and GPS + Galileo + BDS-3 in the ionosphere active period are 1.84, 1.75, and 1.69 cm, respectively, while the corresponding values in the vertical component are 4.88, 4.51, and 4.33 cm. In contrast, during the ionosphere calm period, similar performance is shown, with horizontal accuracies reaching 1.74, 1.62, and 1.51 cm, and vertical accuracies attaining 4.31, 3.92, and 3.53 cm for the aforementioned systems, respectively. However, in the ionospheric medium condition, the horizontal component achieves 1.79, 1.70, and 1.58 cm, and the vertical component achieves 4.47, 4.21, and, 3.97 cm, respectively. Overall, multi-frequency signals significantly enhance positioning accuracy, yet their influence by ionosphere condition is still relatively significant.

When comparing the positioning accuracy of different systems between the ionosphere active and calm periods in both horizontal and vertical components, notable differences in improvement percentages are observed. In the horizontal component, the differences percentages are 5.5%, 7.4%, and 10.4% for GPS-only, GPS + Galileo, and GPS + Galileo + BDS, respectively. In the vertical component, they are 11.8%, 13.0%, and 18.5%. This also indicates that the positioning accuracy of PPP-RTK is significantly affected by ionospheric activities. During the ionospheric active period, both the positioning accuracy and the convergence speed will be affected, resulting in performance degradation. Overall, while multi-frequency signals can enhance positioning accuracy to some extent, the influence of ionosphere conditions remains significant and varies among different systems and between horizontal and vertical components.

Through the experimental tests in this study during the active and clam periods of the ionospheric condition, it can be seen that PPP-RTK has been significantly affected in terms of atmospheric estimation at the server end, UPD calculation, and positioning performance at the user end. During the ionospheric active period, the WL UPD also shows more significant fluctuation changes. The STDs of GPS, Galileo, and BDS satellites increase by 3.3%, 28.6%, and 17.9%, respectively, compared to the ionospheric calm period. In addition, for the narrow-lane UPD sequence, these three systems are affected by the ionosphere active condition, resulting in an increase of 34.3%, 20.6%, and 20.0% in STD. This also leads to a more discrete residual distribution of the ambiguity fixing, thus making the ambiguity fixing unable to be performed at some epochs.

Affected by the deterioration of UPD product quality, the accuracy and success rate of ambiguity fixing performed by each station also decrease. Therefore, the fluctuations of the estimated tropospheric and ionospheric delay products also increase significantly. Compared with the ionospheric calm period, the fluctuation of tropospheric delay significantly increases by 17.3%, while the ionosphere shows a greater change. The fluctuations of the three constellations of GPS, Galileo, and BDS increase from 4.9 to 13.9 cm.

For the positioning, the UPD product affected by ionospheric anomalies also has a great impact on ambiguity fixing. At the same time, the estimated atmospheric products also have large fluctuations and differences. According to the statistical positioning results, compared with the ionospheric calm period, the convergence speed and accuracy of positioning have a large drop. That is to say, for the most mainstream PPP-RTK technology at present, the abnormality of the ionosphere will have a significant impact on its performance. Whether it is the UPD product at the service end or the extracted atmospheric parameters, the final positioning result will be affected by this.

## 6. Conclusions

This study examines the influence of UPD estimation, atmospheric delay derivation, and PPP-RTK under calm and highly active ionospheric conditions. The ionospheric delay and tropospheric ZWD are derived to assess active and calm ionospheric periods. Additionally, the derived atmospheric delays are interpolated from nearby stations and introduced to the PPP-AR model for correction. Finally, positioning errors and convergence times are statistically analyzed. The conclusions are as follows:

During the ionospheric calm period, both UPD estimation and atmospheric extraction exhibited stable performance, allowing PPP-RTK positioning to achieve rapid convergence within 1 min at the 68th percentile. In contrast, during the ionospheric active period, both residuals and the stability of UPD estimation are affected, with WL and NL residuals increasing by 2.4% and 1.4%, respectively. The impacts on the troposphere and ionosphere are significant, with tropospheric differences increasing by 17.3% and ionospheric differences reaching 167.1%.

For user-end positioning, the ionospheric anomalies led to a decrease in PPP-RTK positioning accuracy by 11.6%, 9.0%, and 10.2% in the ENU directions, respectively. The convergence speed also decreased by 203.6%, 183.3%, and 76.7% in the ENU directions, respectively.

In general, we analyzed and compared the influence of ionospheric anomalies on PPP-RTK and the performance differences of PPP-RTK during the ionospheric quiet and active periods through the ionospheric anomaly period in the Australian region. The research and analysis in this paper can be used as a performance reference for PPP-RTK services during the ionospheric anomaly period.

## Figures and Tables

**Figure 1 sensors-25-02200-f001:**
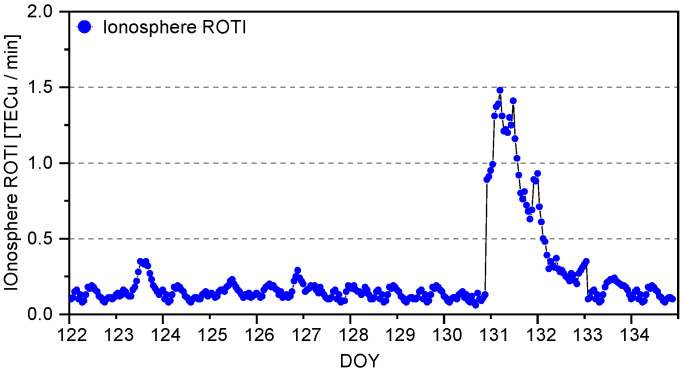
Ionosphere ROTI from 1 to 14 May 2024.

**Figure 2 sensors-25-02200-f002:**
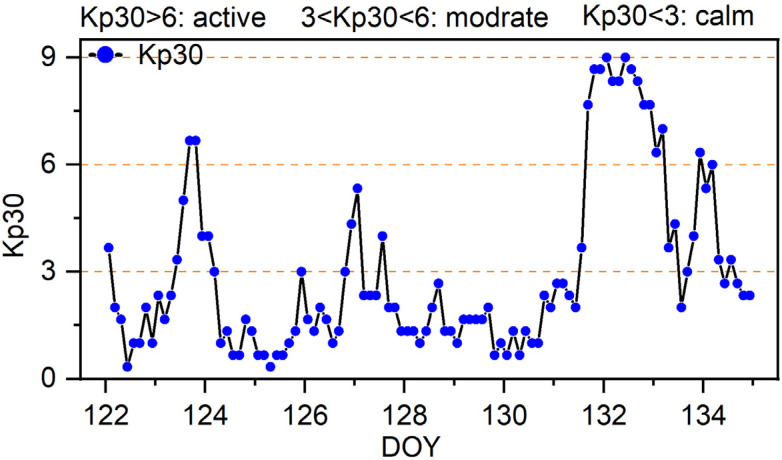
Ionosphere activity index from 1 to 14 May 2024.

**Figure 3 sensors-25-02200-f003:**
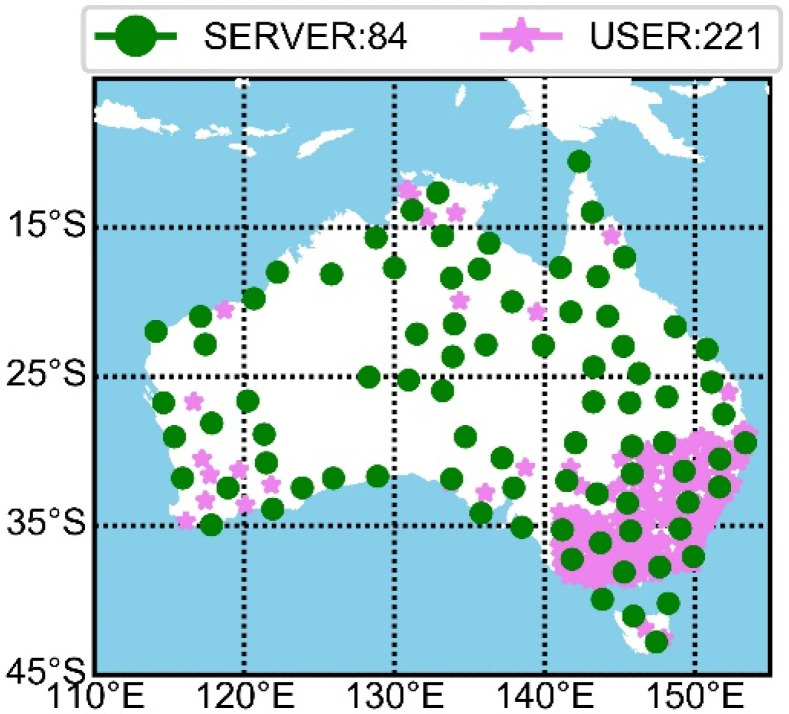
The GNSS networks are used to derive atmospheric delay and for positioning verification. Green dots with 200 km station spacing are service-side stations and pink asterisks are user-side stations.

**Figure 4 sensors-25-02200-f004:**
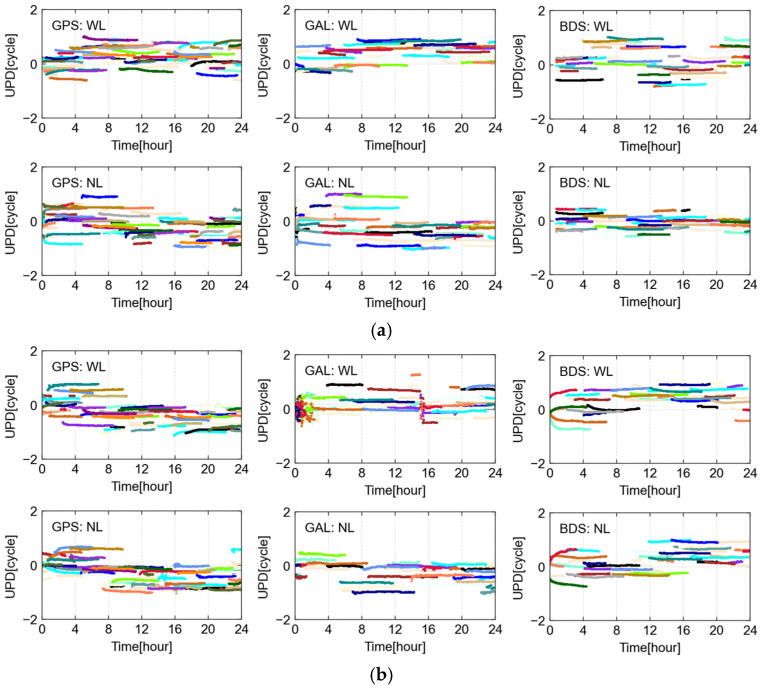
WL and NL UPD estimated in the Australian region. The top panel (**a**) is the ionosphere calm condition on DOY 130, and the bottom panel (**b**) is the ionosphere active condition on DOY 132. Different color lines represent different satellites.

**Figure 5 sensors-25-02200-f005:**
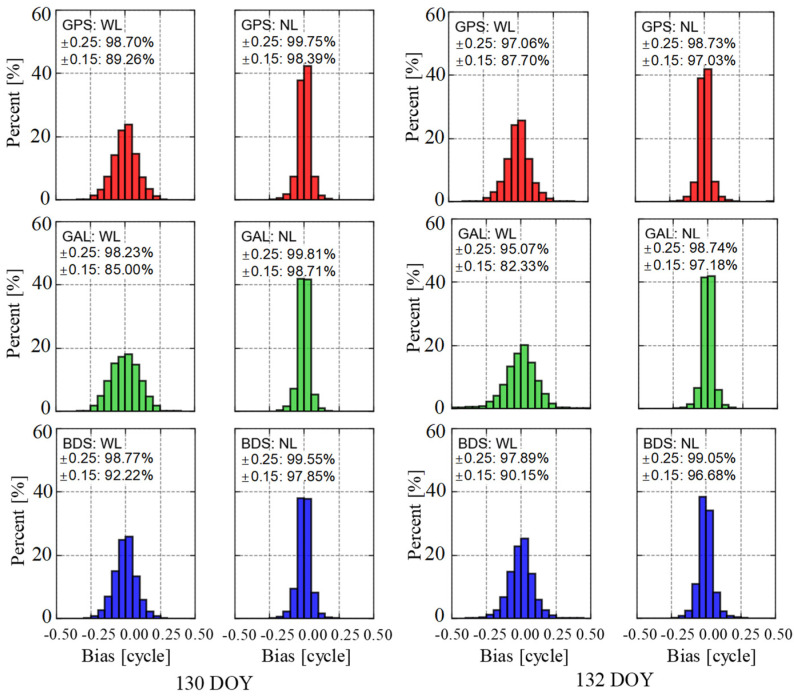
GPS, Galileo, and BDS satellites fractional bias of WL and NL UPDs on different ionosphere conditions. The (**left**) panel is ionosphere calm on DOY 130, and the (**right**) panel is ionosphere active on DOY 132.

**Figure 6 sensors-25-02200-f006:**
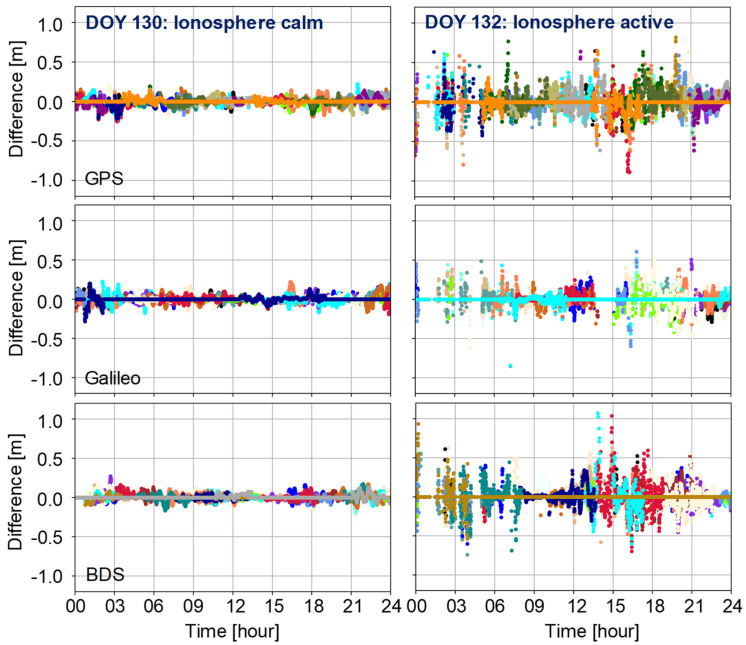
Ionospheric delay difference between PPP-AR-derived and three nearby stations interpolated on CNDO station. The solid lines denote the difference between reference station PPP-AR derived and nearby stations interpolated. Different colors denote different satellites.

**Figure 7 sensors-25-02200-f007:**
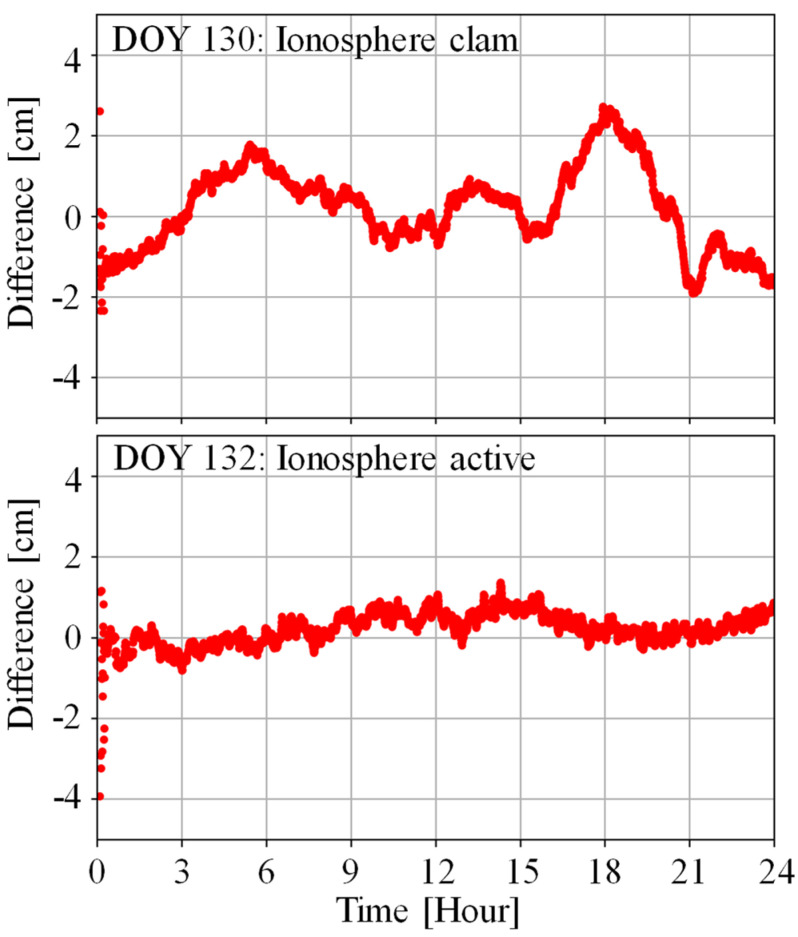
Tropospheric ZWD difference between PPP-AR-derived and nearby station interpolated on CNDO station.

**Figure 8 sensors-25-02200-f008:**
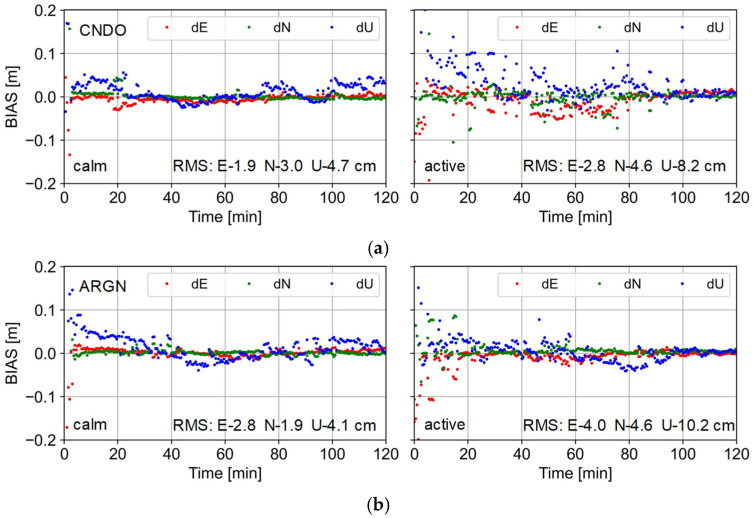
The positioning results on CNDO (**a**) and ARGN (**b**) stations under ionosphere calm (**left**) and active (**right**) conditions.

**Figure 9 sensors-25-02200-f009:**
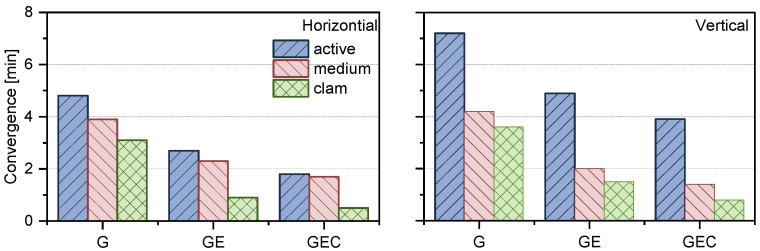
Convergence time of GPS-only, GPS + Galileo, and GPS + Galileo + BDS in different ionosphere conditions.

**Figure 10 sensors-25-02200-f010:**
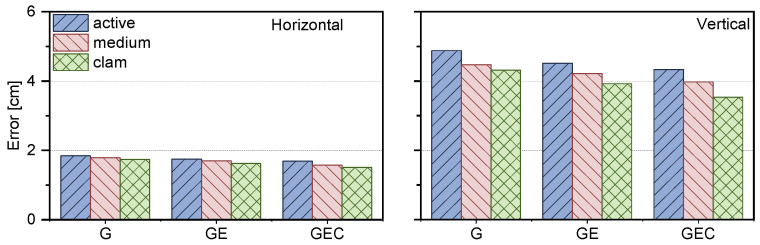
The average result of PPP-RTK positioning accuracy for different constellation combinations.

**Table 2 sensors-25-02200-t002:** Different ionosphere conditions’ tropospheric ZWD and ionospheric delay differences between PPP-AR-derived and nearby stations interpolated. The average values are presented.

Type	Calm [cm]	Medium	Active [cm]
Ionosphere	4.97	7.83	13.86
Troposphere	1.01	1.15	1.32

## Data Availability

The orbit and clock products we used are GFZ real-time products from 139.17.3.115:2101/SSRA00GFZ0. The GNSS data are provided by Geoscience Australia, which can be accessed from https://ga-gnss-data-rinex-v1.s3.amazonaws.com/index.html#public/daily/, accessed on 1 May 2024, and DCB products are from CAS, which is publicly accessed from IGS.

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
