# Peer review of "Multi-GNSS Large Areas PPP-RTK Performance During Ionosphere Anomaly Periods"

_sensors, 2025, doi:10.3390/s25072200_

Round 1
Reviewer 1 Report
Comments and Suggestions for Authors
This study investigates the performance of multi-GNSS precise point positioning with real-time kinematic (PPP-RTK) technology during ionospheric anomaly periods. The research analyzes 13-day data from 305 Australian stations, covering both ionospheric anomalies and calm periods. The study evaluates the residuals of uncalibrated phase delay (UPD), the accuracy of atmospheric modeling, and the positioning accuracy and convergence time of PPP-RTK. The paper is innovative and can be accepted for publication after a major revision.
Major
The study uses 14 days of observation data and classifies ionospheric activity as active, modrate, and calm. The authors should analyze the ambiguity estimation and positioning performance for the two ionospheric conditions (modrate and active) separately. The authors should compare the performance in both conditions relative to the performance in the calm state.
Minor
Line 138. The spaces in front of word ‘where’ should be deleted, there are many similar problems
Line 170. The font of the paper seems to be inconsistent.
Reviewer 2 Report
Comments and Suggestions for Authors
Article entitled ‘Multi-GNSS Large areas PPP-RTK performance during ionosphere anomaly periods' presents the results of very interesting research related to the influence of the ionosphere on GNSS positioning results. However, it requires improvement:
1. Line 41-42: The sentence is somewhat unclear.
2. Line 141: notation: ‘all necessary corrections listed in Error! Reference source not found.’ is not clear.
3. Formula (9): Please check the correctness of the notation.
4. Please standardise the format of tables throughout the article.
5. Some item references seem too old to cite in the article.
Reviewer 3 Report
Comments and Suggestions for Authors
This manuscript investigates the positioning performance of PPP-RTK (Precise Point Positioning with Real-Time Kinematic) during periods of "ionospheric disturbance" using undifferenced and uncombined methods, and evaluates their impact, which is somewhat significant in terms of its results. The authors emphasize the importance of ionospheric anomalies, however, the manuscript lacks a comprehensive description of the data related to these anomalies. It is essential to note that the Kp index is a scale that measures global geomagnetic activity within the Earth's magnetosphere, ranging from 0 to 9, where higher values indicate increased geomagnetic activity. This index is derived from magnetic field fluctuations recorded at various ground-based observatories worldwide. However, it is critical to clarify that the geomagnetic Kp index does not equate to an index of ionospheric disturbance. The authors incorrectly utilize the Kp index to define the extent of ionospheric disturbances. While significant changes in the global ionosphere occur during geomagnetic storms, particularly during major storms, the ionosphere's response to magnetic storms is inherently complex and involves various mechanisms that vary by region. This is the primary issue with the manuscript.
Main Points:
- The ionospheric disturbance should be characterized using established indices such as the Rate of Total Electron Content Index (ROTI) or the S4 index.
- The term "abnormal ionospheric fluctuations" requires definition and elucidation; how can these fluctuations be quantified or represented?
- The manuscript references the Kp30 index in Figure 1; what exactly is the Kp30 index? How does it differ from the Kp index, and what are the sources of these indices?
- In Figure 2, the server-side stations can serve as reference stations for the precise estimation of ionospheric Total Electron Content (TEC) using traditional geometry-free linear combination methods. Why, then, is the ionospheric delay treated as white noise for estimation purposes?
- It is important to recognize that ionospheric delay anomalies are merely surface phenomena. The fundamental cause of degraded positioning performance during ionospheric disturbances is likely due to frequent cycle slips, loss of lock, or interruptions in navigation signals, which directly affect the estimation of UPD, ambiguity resolution, and convergence time. These aspects should be thoroughly researched and discussed in the manuscript.
- PPP-RTK technology does not solely rely on undifferenced and uncombined methods. Although these methods are widely applied in PPP-RTK, they are not the exclusive option. Have the authors considered the potential of using ionosphere-free combinations to mitigate the effects of ionospheric anomalies?
Round 2
Reviewer 1 Report
Comments and Suggestions for Authors
I'm not satisfied with the author's response to the first comment. In the previous version of the manuscript, the author classified the ionospheric activities into three categories: active, moderate, and calm. Why is it now directly divided into only two stages, where is the moderate stage?
The innovative points of this manuscript lie in the refined classification of the active stage of the ionosphere and the assessment of the positioning performance during different stages. According to the current revisions, the innovative points of the paper are not sufficient for publication in this journal. It is recommended that the author continue to make revisions.
Comments on the Quality of English Language
no problems.
Author Response
We appreciate your comments and suggestions on our manuscript. We are grateful for your meticulous and careful comments on our main text and figures.
Also, many thanks for the reviewers’ helpful comments on our previous submission. The manuscript has been revised following the reviewers’ detailed comments. We also try the best to further improve the language clarity and minimize the typo, syntax and grammar errors. We have highlighted the major modifications in the revised manuscript for easy reading. The reviewers’ comments, as listed below, are carefully responded to in sequence.
The original comments of the reviewer are cited in italic font, and our response is listed below each comment in the color of blue.

Reviewer 3 Report
Comments and Suggestions for Authors
The ROTI index and the ionospheric scintillation index are strongly correlated and are often used as a proxy for the strength of ionospheric scintillation due to ionospheric irregularities. The results on the ROTI index in the revised manuscript appear to differ from the traditional results. It is unlikely that the traditional 5-minute ROTI index would also show high levels during daytime. The suggested modifications are as follows:
- For the ROTI index given in Figure 1, please explain what data were used and what methodology. How did the authors get the regional ROTI index over Australia?
- How were Figures 2 and 3 in the Coverletter document obtained? What data were used?
- Also, in the reply to comment 4 of the coverletter file, the authors added the Kp index, but the Kp index and its description do not appear in the revised manuscript.
Author Response

(The authors gave the same response as above.)

Round 3
Reviewer 1 Report
Comments and Suggestions for Authors
Agreeing to the publication of papers